# Evaluation of general anesthesia protocols for a highly controlled cardiac ischemia-reperfusion model in mice

**Christelle Leon[1,2], Alice Ruelle[1,2], Juliette Geoffray[1,2], Lionel Augeul[1,2], Catherine Vogt[3], Pascal Chiari[1,4], Ludovic Gomez[1,2], Michel Ovize[1,2], Gabriel Bidaux[1,2], Bruno Pillot[1,2]***

**1** Université-Lyon, CarMeN Laboratory, Inserm U1060, INRA U1397, Université Claude Bernard Lyon 1, Bron, France, **2** Département de Cardiologie, Hospices Civils de Lyon, Groupement Hospitalier EST, IHU-OPERA Bâtiment B13, Bron, France, **3** Université-Lyon, Ecole de Chirurgie et de Gestes Micro-Invasifs de Lyon, Université Claude Bernard Lyon 1, Lyon, France, **4** Service d'Anesthésie-Réanimation, Hôpital Louis Pradel, Hospices Civils de Lyon, Lyon, France

\* bruno.pillot@univ-lyon1.fr

## Abstract

### Background

The aim of our study was to test different anesthetic mixtures in order to identify the most suitable one for a surgical cardiac ischemia-reperfusion model in mice.

### Methods

1) Sixty four mice were submitted to one of the 6 combinations of ketamine or alfaxalone associated to xylazine, medetomidine or midazolam. Depth and quality of anesthesia were evaluated via 5 reflex scores. 2) Impact of analgesic (buprenorphine or butorphanol), anesthesia reversal (with atipamezole) and surgery (cardiac ischemia-reperfusion surgery) have been tested in the selected protocols. 3) infarction size has been measured with TTC (Triphenyl Tetrazolium Chloride) method in mice anesthetized with best protocols.

### Results

Protocol involving medetomidine induced the longest surgical anesthesia: (median = 120, {interquartile range = 100–125}) min with ketamine and 53 {25–100} min with alfaxalone. Butorphanol substitution with buprenorphine did not alter time-related anesthesia parameters. Atipamezole reversal considerably reduced both recovery and immobilization time (respectively 22 {18–30} min and 98 {88–99} min vs. 55 {40–70} min and 143 {131–149} min, in groups with no reversal, p = 0.001) with no impact on infarction size measurement.

### Conclusion

In this study, the combination alfaxalone/medetomidine/buprenorphine (80/0,3/0,075 mg. kg⁻¹, s.c.) associated with reversal by atipamezole was a reliable anesthetic protocol for murine surgery, particularly for the study of ischemia-reperfusion.

**Data Availability Statement:** All relevant data are within the manuscript and its Supporting Information files.

**Funding:** The author(s) received no specific funding for this work.

**Competing interests:** The authors have declared that no competing interests exist.

**Abbreviations:** TTC, Triphenyl Tetrazolium Chloride; NMDA, N-methyl-D-aspartate; GABA, γ-aminobutyric acid A; EtCO2, End-tidal Carbon dioxide; NIP, NonInvasive Pressure; ECG, ElectroCardioGram; LAD, Left Anterior Descending; S.D, Standard Deviation; BT, Body Temperature; BP, Blood Pressure; HR, Heart Rate; min, minutes.

## Introduction

For the last decades, intraperitoneal pentorbabital sodium injection was commonly used for anesthesia in mice as well as in other animal species because of his efficiency, stability and reproducibility in terms of duration and depth of anesthesia. Nevertheless, it was gradually replaced by more complex mixture allowing a wider analgesic coverage and a better recovery of the animal, with different administration routes. However, anesthesia can induce interferences on experimental models, therefore a better characterization of anesthetics must be achieved to prevent confounding factors in the analysis [1].

Model of cardiac ischemia-reperfusion in rodents relies on a surgical procedure under anesthesia and analgesia in order to immobilize the animal and reduce its stress and pain. Most of recent protocols of anesthesia have consisted in a mixture of several anesthetic and analgesic agents allowing better sedation, narcosis, myorelaxation and analgesia of animals [2–4]. However, each of these anesthetic protocols differentially modify physiological parameters, and in particular the cardiac function in rodents [5,6], and thus could interfere in the analysis of experimental data. The small size of a mouse can compromise the efficiency of anesthesia because of 1) logistical difficulties to precisely monitor depth of anesthesia, 2) the risk of overdose, and 3) the risk of hypothermia.

Nowadays, volatile anesthetics are often preferred for rodents, especially for brief procedures, due to their reliability and ability to achieve both rapid induction and recovery [7]. However, volatile anesthetics also exhibit undesirable effects (on blood pressure and heart rate) and even confounding effects complexifying data interpretation (i.e. isoflurane or sevoflurane are cardioprotective [8,9]). For these reasons, liquid anesthetics are still useful and their effects on physiological parameters must be investigated carefully.

Surgical anesthesia is always complemented by one or several analgesic agent(s) in order to properly suppress any pain due to experiment. Since analgesia effects can interact with anesthesia effects and generate potential confounding effects, a careful characterization of analgesic and anesthetic agents should be provided [10,11]. Multiple analgesia pathways can be combined to correctly prevent surgery-induced pain. In addition of local anesthetic, the centrally-acting analgesic butorphanol and buprenorphine are commonly used for the treatment of moderate to severe pain du to surgery [12,13].

The aim of our study was to compare several combinations of anesthetics and analgesics in terms of efficacy and reduction of side effects on the measurement of infarction size in our mouse model of cardiac ischemia-reperfusion. Anesthesia induction and sedation were measured in mice anesthetized with ketamine or alfaxalone associated to: Xylazine, Medetomidine or Midazolam. Besides, the effects of butorphanol and buprenorphine on anesthesia settings were compared. We also assessed the reversal time of anesthesia with the α2-adrenergic receptor antagonist atipamezole, administered in order to accelerate and improve awakening. To evaluate if the surgical procedure itself could also modify both duration and quality of anesthesia, efficacy of anesthesia and awakening was figured out in a mouse model of cardiac ischemia-reperfusion. Finally, we measured the potential side effects of the best anesthetic/analgesic mixture on infarction size in mice subjected to cardiac ischemia-reperfusion.

## Methods

### Animal model and surgery

Male C57bl/6J mice (8–20 weeks old weighing between 22 and 31 g–obtained from Charles River laboratories, France) were studied. Such as other studies working on the cardiac ischemia-reperfusion model, we focused this research only on male mice because of the potentially

cardioprotective effect of estrogens [14,15]. This also reduce the total number of animals used by overcoming the difference between sexes. Animals were housed in stable groups of four in individually ventilated cages (Nextgen—Allentown, USA–conventional animal facility) with standard nesting materials (cotton, tunnel) and ad libitum access to filtered water and standard diet (2018 global rodent diet, Envigo, France). Room temperature (housing and experiment) was maintained at 22˚C ± 2˚C and light cycle was at 12:12.

Animal procedures were performed in accordance with the guidelines from Directive 2010/ 63/EU on the protection of animals used for scientific purposes and have been approved by the institutional animal research ethical committee from Université Claude Bernard Lyon 1 and French ministry (authorizations APAFIS#9038–2017022407585959, APAFIS#10333– 2017062220074257 and APAFIS#29506–2021020414463033).

Study design was performed according to PREPARE guidelines and experimentations were performed according to the ARRIVE guidelines.

The number of animals required in each experiment was determined a priori by power tests calculated with G*Power 3.1 software (one-way or two-way ANOVA with alpha: 0.05 / power 80% / effect size 0.5): N = 12 mice needed for the comparative studies of anesthetics and analgesic protocols on anesthesia settings; N = 10 for the studies concerning the effects of surgery on anesthesia settings and concerning the effects of atipamezole on anesthesia settings and infarction size.

## Anesthetic protocols

Drugs were extemporaneously diluted in saline (0.9%) to prepare injectable solutions at the concentrations indicated in Table 1.

All experiments were performed in the morning to limit perturbation induced by the circadian rhythm [16]. With respect to the study of anesthesia parameters of anesthetic and analgesic agents, mice were allocated by random draw to any of the anesthetic protocols (one per mice and 12 mice per protocol). Mice were weighted (to determine the exact anesthetic volume/concentration injection) and held in the hand to injected with anesthetic and analgesic agents (intra-peritoneal or subcutaneous injection in the lower right dial of the abdomen). After injection mice were placed back to their cage until the righting reflex was lost. Then, mice were placed in dorsal recumbency on a retro-regulated heating pad (target at 37.5˚C–

Table 1. Characteristics of drugs and parameters of administration.

| Drug | Substance type | Effect | Injection concentration | Administration mode | Dilution |
|---|---|---|---|---|---|
| Ketamine | NMDA (N-methyl-D-aspartatereceptor) antagonist | Dissociative ++, Sedation, analgesia | 100 mg.kg$^{-1}$ | Intra-peritoneal | 1/10e |
| Alfaxalone | GABA (γ-aminobutyric acid A) A receptor modulator | Neurodepression ++, myorelaxation | 80 mg.kg$^{-1}$ | Subcutaneous | - |
| Xylazine | α2 adrenergic agonist | Sedation ++, analgesia, myorelaxation ++ | 5 mg.kg$^{-1}$ | Intra-peritoneal | 1/20e |
| Medetomidine | α2 adrenergic agonist | Sedation ++, analgesia, myorelaxation ++ | 0.3 mg.kg$^{-1}$ | Subcutaneous | 1/17e |
| Midazolam | Benzodiazepine, GABA receptor agonist | Sedation++, myorelaxation ++ | 4 mg.kg$^{-1}$ | Subcutaneous | 1/20e |
| Atipamezole | α2 antagonist | Antidote to sedation | 0.3mg.kg$^{-1}$ | Subcutaneous | 1/80e |
| Buprenorphine | Mu opioid partial agonist / Kappa-Mu opioid receptor antagonist | Analgesia +++ | 0.075 mg.kg$^{-1}$ | Subcutaneous | 1/20e |
| Butorphanol | Mu-kappa opioid receptor partial agonist / Mu opioid receptor antagonist. | Analgesia ++ | 5 mg.kg$^{-1}$ | Subcutaneous | 1/10e |

Physiosuite, Kent scientific, USA). Mice were breathing 30%$O_2$-enriched using a mask connected to a system involving air/O2 bottles and adjustable gas mixer and flowmeter. The initial study involved an air supply (21% O2), but such a condition caused unexpected mortality from the first experiments (described further). For ethical reasons and in accordance with the animal welfare structure, the project was stopped and resumed with 30% O2 for all animals. Previous animals with 21% O2 supply are therefore not included in this study.

With respect to the study of surgery and anesthesia reversal impact on anesthesia parameters and infarction size, three new groups (n = 10 per condition) received the previously selected anesthetic protocol (alfaxalone/medetomidine/buprenorphine; 80/0,3/0,075 mg.kg-1, s.c.), two of which also received atipamezole injection (with or without surgery). Atipamezole was administrated 70 minutes after anesthetic induction, a delay shorter enough to occur before the recovery time (determined in the group with no anesthesia reversal). All anesthetic protocols were completed with opioids (butorphanol or buprenorphine).

## Drugs

Medetomidine (Medetor 1mg.ml$^{-1}$, Alcyon, France), ketamine (Imalgen 100 mg.ml$^{-1}$Alcyon, France), xylazine (Rompun 20 mg.ml$^{-1}$, Alcyon, France), alfaxalone (alfaxalone 10 mg.ml$^{-1}$, Alcyon, France), buprenorphine (Vetergesic 0.3 mg.ml$^{-1}$, Alcyon, France), butorphanol (Butador 10 mg.ml$^{-1}$, Alcyon, France) atipamezole (Atipam 5 mg.ml$^{-1}$, Alcyon, France).

## Depth of anesthesia evaluation by scoring of reflexes

Five reflex tests were carried out on animals to evaluate depth and duration of anesthesia, every 10 minutes (surgery group) or 5 minutes (other groups) after loss of righting reflex [17] (Table 2).

## Time-related parameters of anesthesia

Loss of righting reflex time was time from injection of the anesthetic agent until the loss of the righting reflex. Induction time is defined as the time span between loss of right reflex and the start of surgical anesthesia. Surgical anesthesia time is defined as the time during which the score is maximum. Recovery time is defined as the time from the end of surgical anesthesia until the righting reflex returns. Immobilization time is defined as the time span between the

**Table 2. Reflex scoring test.**

| Reflex | Score = 0 | Score = 1 |
|---|---|---|
| **Righting reflex** | Attempt of animal to right itself | no reaction |
| **Tail pinch reflex** | Any reaction | no reaction |
| **Pedal withdrawal reflex of fore legs** | Attemps of withdraw any limb | no reaction |
| **Pedal withdrawal reflex or hind legs** | Attemps of withdraw any limb | no reaction |
| **Palpebral reflex (air is blown on the eyes of the animal through a 5 ml syringe)** | Eyelid reaction | no reaction |

The total score was the sum of all five reflexes. A satisfying surgical-level anesthesia was reached when the total score was maximum.

loss and recovery of righting reflex. Immobilization time is the sum of induction time, surgical anesthesia time and recovery time (Table 2).

### Ischemia-reperfusion experimental procedure

Mice were anesthetized with alfaxalone+medetomidine+buprenorphine (80/0,3/0,075 mg.kg-1, s.c) and placed on retro-regulated heating pad as previously described in "Anesthetic protocols". Lidocaine at 1 mg.kg$^{-1}$ (Laocaine 20 mg.ml$^{-1}$, Alcyon, Civrieux, France), a local anesthetic was injected subcutaneously at the incisional areas. Animals were intubated and ventilated with Physiosuite ventilator (Kent scientific, USA) whose breath rate and tidal volume are automatically adapted to the weight of animal. Body temperature, pad temperature and end-tidal carbon dioxide (EtCO2) were continuously monitored thanks sensors (temperature probes and capnograph) connected to the Physiosuite ventilator. The measurement sleeve and inflation cuff of non-invasive Pressure (NIP) CODA system (Kent scientific, USA) were placed on animal tail to measure systolic blood pressure during the immobilization time. Subcutaneous needles of electrocardiogram (ECG) monitoring system (Emka, Paris, France) were placed at the right forearm, right leg and left leg. A left thoracotomy was performed and left anterior descending artery (LAD) occlusion was performed with an 8–0 polypropylene nylon suture. After 60 minutes of ischemia, reperfusion was allowed by loosening the snare loop around the LAD. Wounds on thoracic cage and skin were sutured with 5–0 cardioxyl suture and endotracheal tube was removed once spontaneous breathing resumed. Air in the thoracic cavity is sucked through a catheter connected to a syringe during suturing to prevent pneumothorax. Animals received 300μL NaCl (37˚C) subcutaneous injection to prevent dehydration and were placed into a heating incubator (Temsega, Pessac, France) until total recovery that is to say up to recovery of normal behavior and reaction to stimuli. Upon waking, in addition of usual food and bottle of water, gelled food and gelled water are given to the animal to facilitate food intake. Postoperative pain evaluation was performed at different times and was based on clinical symptoms, mainly respiratory, behavioral and morphological. Analgesia was maintained until the end of experiment by injection of Buprenorphine (0.075 mg.kg$^{-1}$) administered every 6–8 hours, according to postoperative pain scoring.

At 24h of reperfusion, the area at risk and the area of necrosis were determined by methods previously describes [18]. Mice were deeply anesthetized and the coronary artery was briefly reoccluded before intravenous injection of unisperse blue to delineate the in vivo area at risk. Hearts were excised and left ventricle was cut into 5 transverse slices (1.5 mm), parallel to the atrioventricular groove after atrial and right ventricular tissues were trimmed off. The basal surface of each slice was photographed for later measurement of the area-at-risk. Each slice was then incubated for 10 minutes in a 1% solution of triphenyltetrazolium chloride at 37˚C. This method has been shown to reliably identify area of necrosis (which appears pale) from viable myocardium (which stains brick red) (**Fig 1**). The slices were weighed and rephotographed. Area-at-risk and area of necrosis extent were quantified by computerized planimetry and corrected for the weight of the tissue slices. Total weights of the area-at-risk and area of necrosis were then calculated and expressed in grams and as percentages of total left ventricle, or of the area-at-risk weight, respectively.

### Statistical analysis

Results are expressed as median {interquartile ranges} (Figs 2–4) or mean ± standard deviation (Table 3). The data were blindly analyzed with R software (for statistical analysis) and graphs plotted using GraphPad Software (Inc, SanDiego, USA).

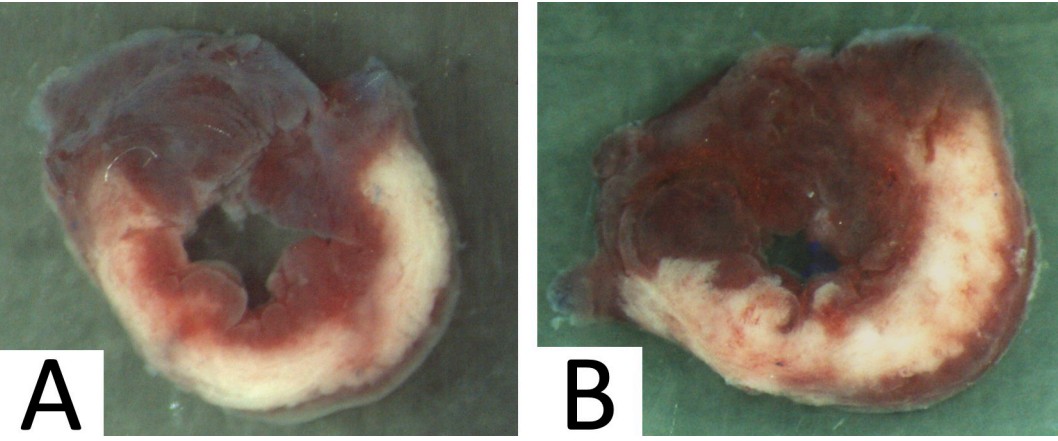

**Fig 1. Infarction visualization.** Representative images of TTC staining obtained at mid left ventricular level after 60 minutes of ischemia followed by 24 hours after ischemia-reperfusion. Viable myocardium is colored red, whereas infarcted myocardium appears pale. A, mouse heart of control group B, mouse heart of group with atipamezole.

Statistical considerations: data distribution and homoscedasticity were determined for each data set to choose appropriate statistic tests (with a Shapiro test or a Levene test).

Fig 2: for each of the 6 time-related parameters of anesthesia, protocols were compared by Kruskal-Wallis followed by a pair wise comparison with a Dunn test where the p-values were adjusted by Bonferroni method.

Fig 3A–3D: for each of the 6 time-related parameters of anesthesia, protocols were compared by Wilcoxon test.

Fig 4: data were analyzed by ANCOVA (measurement of the covariance of least squares fitting for the two lines with ratio of the two slopes).

Table 3: data were analyzed by ANOVA1 (Friedman test) followed by a t-test for the pair wise comparison, adjusted by the Tukey method.

## Results

### Comparative effects of anesthetic protocols on anesthesia settings

Six different anesthetic protocols were tested, combining alflaxalone or ketamine, with xylazine, midazolam or medetomidine. An analgesic agent, the butorphanol was added to every

**Table 3. Hemodynamic parameters monitored during surgery.**

| Groups | Ctrl group | | | | | | | | | Surgery group | | | | | | | | | |
|---|---|---|---|---|---|---|---|---|---|---|---|---|---|---|---|---|---|---|---|
| Timepoints | 10 minutes of anesthesia | | | During ischemia | | | After reperfusion | | | 10 minutes of anesthesia | | | During ischemia | | | | After reperfusion | | |
| HR (bpm) | 380 | ± | 49 | 352 | ± | 55.33 | 334 | ± | 43 | 358 | ± | 27 | 286 | ± | 50 | | 247 | ± | 61 | |
| BP (mm Hg) | 94 | ± | 28 | 83 | ± | 19 | 89 | ± | 23 | 104 | ± | 30 | 82 | ± | 19 | | 76 | ± | 19 | * |
| Body temperature (˚C) | 37.1 | ± | 0.9 | 37.2 | ± | 0.1 | 37.3 | ± | 0.3 | 36.7 | ± | 0.6 | 37.1 | ± | 0.2 | | 37.3 | ± | 0.1 | *$ |
| Pad temperature (˚C) | 37.8 | ± | 3.2 | 36.0 | ± | 2.2 | 35.4 | ± | 2.7 | 38.7 | ± | 1.9 | 37.6 | ± | 1.6 | * | 36.7 | ± | 2.2 | * |
| EtCO2 (mmHg) | NC | | | NC | | | NC | | | 16.8 | ± | 3.2 | 18.3 | ± | 4.0 | | 16.9 | ± | 2.2 | |

Data are presented as means ± SD. For each parameter, differences between values of the 3 timepoints of a same group (ctrl or surgery) were analyzed by ANOVA1 test followed by a t-test, adjusted by Tukey method. P value < 0.05 is considered to be statistically significant

* p<0.05 compared to "10min of anesthesia" value and

$ p<0.05 compared to "During ischemia" value. N = 7–10 according parameter.

**Fig 2. Characterization of 6 different anesthetic protocols.** A. Combinations of the different anesthetics and analgesics. B. Depth of anesthesia and time-related parameters were calculated with the reflex test scores. Protocol 6 resulted in a high mortality rate and was therefore stopped before ending the procedure. Differences between groups were analyzed by Kruskall-Wallis followed by a pair Dunn post-hoc test adjusted by Bonferroni method. Data are presented as median and interquartile ranges. P value < 0.05 is considered to be statistically significant. Numbers indicate that the protocol below is significantly different (p<0.05) from the protocol sharing the same number. N = 12 per group.

protocol. As reported in the **Fig 2**, righting reflex was lost in less than 3 minutes in all groups and the induction time of anesthesia was almost similar among the different protocols (respectively 28 {19–34} min, 13 {0–19} min, 19 {19–24} min, 21 {14–28} min and 12 {9–14} min protocols 1, 2, 3, 4 and 5). For each time-related parameter of anesthesia measured, protocols were compared to each other.

In protocol 1, a mixture of ketamine (100 mg.kg$^{-1}$), xylazine (5 mg.kg$^{-1}$) and butorphanol (5 mg.kg$^{-1}$) was used. No mortality was observed. Two animals never reached surgical stage of anesthesia (= score 5). Protocol 2 (alfaxalone 80 mg.kg$^{-1}$ + xylazine 5 mg.kg$^{-1}$ + butorphanol 5 mg.kg$^{-1}$) and protocol 5 (ketamine 100 mg.kg$^{-1}$ + midazolam 4 mg.kg$^{-1}$ + butorphanol 5 mg.kg$^{-1}$) both allowed short-to-medium time of anesthesia with respectively 8 {0–36} min and 30 {25–35} min of surgical stage of anesthesia. Five animals never reached surgical stage of anesthesia (= score 5) in protocol 2. In protocol 3 and 4, medetomidine (0.3 mg.kg$^{-1}$) and butorphanol 5 mg.kg$^{-1}$ were respectively used in association with ketamine (100 mg.kg$^{-1}$) or alfaxalone (80 mg.

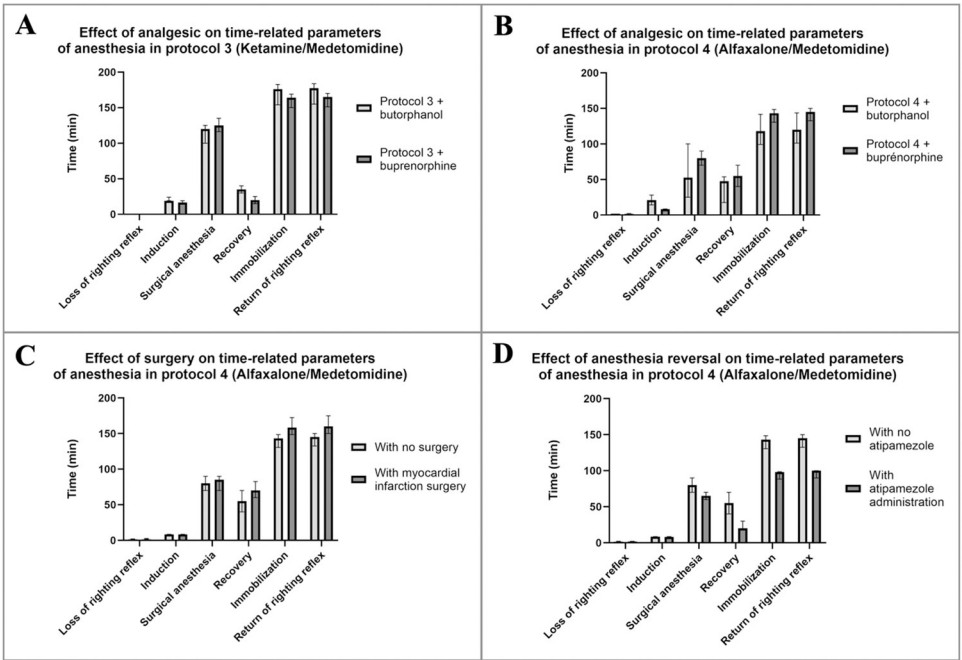

**Fig 3. Effects of two opioids on time-related parameters of anesthesia.** Comparison of two analgesic agents: butorphanol (black bar) and buprenorphine (grey bar) in protocol 3 (**A**) and 4 (**B**) as described in the Fig 2. Depth of anesthesia and time-related parameters were determined by reflex test scores. Data are presented as median and interquartile ranges. Differences between groups were analyzed by Wilcoxon test. P value < 0.05 is considered to be statistically significant.* p<0.05 and ** p<0.01 compared to equivalent anesthetic protocol with butorphanol. N = 12 for butorphanol groups. N = 12 for buprenorphine groups. C. Quality of anesthesia with or without surgery of cardiac ischemia-reperfusion. Depth of anesthesia and time-related parameters were determined by score of reflex tests. Data are presented as median and interquartile ranges. Differences between groups were analyzed by Wilcoxon test. P value < 0.05 is considered to be statistically significant.* p<0.05 and ** p<0.01 compared to group without surgery. N = 12 without surgery and N = 10 with surgery. D. Effects of atipamezole on awakening of anesthetized mice. Time-related parameters were determined by score of reflex tests. Data are presented as median and interquartile ranges. Differences between groups were analyzed by Wilcoxon test. P value < 0.05 is considered to be statistically significant. ** p<0.01 compared to group without atipamezole. N = 12 without atipamezole and N = 10 with atipamezole.

$kg^{-1}$) and surgery stage of anesthesia lasted respectively 120 {100–125} min (significantly higher than protocol 1, 2 and 5 with a p-value <0.01) and 53 {25–100} min. Those protocols are better suited to long-term surgery (about 60min). In protocol 6, despite a O2-enriched air supply (with a mask), the mixture of alfaxalone (80 mg.$kg^{-1}$), midazolam (4 mg.$kg^{-1}$) and butorphanol 5 mg.$kg^{-1}$ induced high mortality (2 mice on 4 tested) that was potentially due to cardiorespiratory failures. This protocol has therefore been stopped for ethical reasons.

Consequently, protocols 3 (ketamine/medetomidine) + butorphanol and 4 (alfaxalone/medetomidine) + butorphanol was judged the most suitable to perform a cardiac ischemia-reperfusion surgery in mice because initial dose of anesthetic agents induced a prolonged anesthesia over the whole surgery sequence. Moreover, anesthesia including medetomidine can be reversed by subsequently administering α2-antagonist to allow a quick recovery.

We next assessed the influence of changing opioid on the anesthetic efficacy and duration of protocols 3 et 4.

## Comparative effects of two opioids on anesthesia settings

The depth of anesthesia and time-related parameters determined in mice anesthetized with Protocol3 (ketamine/medetomidine) + butorphanol or protocol4 (alfaxalone/

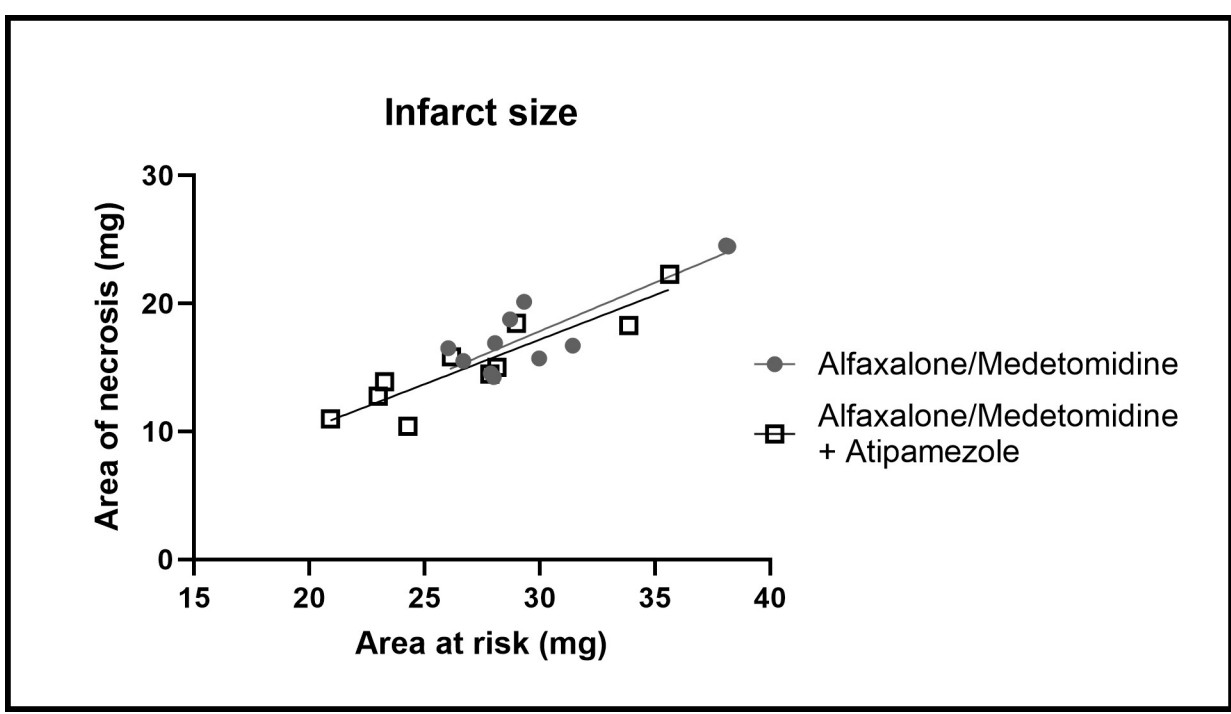

**Fig 4. Effect of anesthesia induction and reversal on infarction size.** Infarction size measured after cardiac ischemia-reperfusion. Data represented the infarction size normalized by the size of area-at-risk (mg of tissue). Each point represents a mouse. Linear fits with least square regression constrained to 0;0 (x;y) were set and the difference of slopes were tested (ANCOVA; GraphPad Prism). P = 0.33.

medetomidine) + Butorphanol in the **Fig 2** were compared to protocol3 + buprenorphine or protocol4 + buprenorphine groups respectively (**Fig 3A and 3B**).

Induction time was slightly reduced in groups with buprenorphine compared to groups with butorphanol (whatever the considered anesthetic protocol). Surgical anesthesia, and immobilization time, were similar whatever the analgesic used in combination of protocol 3. However, for the protocol 4, surgical anesthesia tended to last longer in buprenorphine group compared to butorphanol group, providing a more comfortable time for surgery (80 {70–90} min vs. 53 {25–100} min) but leading to significantly higher immobilization time (143 {131–149} min vs. 118 {99–142} min, p-value <0.01). In summary, the change of analgesic agent did not cause any major alteration of anesthesia settings.

The ketamine/medetomidine protocol has been widely used but paradoxical effects have been reported in the literature [19,20]. In addition, because ketamine has been added to the narcotic list, many laboratories have already turned to alternative anesthetics such as alfaxalone. We thus decided to focus our study on the alfaxalone/medetomidine protocol.

### Validation of alfaxalone/medetomidine protocol in a mouse model of cardiac ischemia-reperfusion

One group of mice was submitted to cardiac ischemia-reperfusion surgery with thoracotomy (total procedure duration was about 70 minutes). Reflex tests were then performed while body and heating pad temperatures (BT and HPT), heart rate (HR), non-invasive systolic blood pressure (BP) and EtCO2 were monitored. Due to HR, BP, and temperatures variations, final values were means of values measured during the 5 last minutes of the phase (for "10 minutes of anesthesia" timepoint), during the whole phase (for "During ischemia" timepoint) or during the 5 first minutes of the phase (for the "after reperfusion" timepoint). Anesthesia and

analgesia used in both groups was those previously selected (alfaxalone/medetomidine/buprenorphine).

Time-related parameters of anesthesia obtained here were compared to those obtained in group without surgery (Fig 3).

Recovery time, and thus immobilization time and return of righting reflex time, were slightly increased in animals undergoing cardiac ischemia-reperfusion surgery, compared to group with no surgery (respectively 70 {60–83} min vs. 55 {40–70} min for recovery time, 159 {149–173} min vs. 143 {131–149} min immobilization time, 160 {150–175} min vs. 145 {133–150} min for return of righting reflex time; Fig 3C). Those data confirmed that surgery procedure did not adversely affect depth and duration of anesthesia.

For each hemodynamic parameter, the three timepoints of a same group were compared (Table 3).

EtCO2 measurement on control group (without surgery) was not reported, because mice were not intubated in this group.

As reported in **the table**, we observed low heart rate and systolic blood pressure (respectively 380 ± 49 bpm and 94 ± 28 mmHg) from anesthesia outset (10min after induction) in control group, compared to admit average physiological values of non-anesthetized mice (around 600 bpm and 120 mmHg respectively) [21].

Moreover, our data showed that HR and BP decreased steadily throughout the procedure in both control group and surgery group (Friedman test p = 0.029 and p = 0.015 respectively). However, no significant difference was observed when compared pairwise, probably because of the limited number of mice per groups. In our surgery group, measurement after 10 minutes of anesthesia was used as basal value before surgery. Our data showed that BP decreased throughout the surgery (76 ± 19 mmHg after reperfusion, compared to 104 ± 30 mmHg before surgery p<0.05). HR did not significantly changed (286 ± 50 bpm during ischemia and 247 ± 61 bpm after reperfusion, compared to 358 ± 27 bpm before surgery). Body temperature was correctly maintained at approximately 37°C during the experiment thanks to the retro-regulated heating pad that adapts the temperature of the pad to maintain the temperature of the animal. Temperature of heating pad tends to decrease over time in the control group and was significant in the surgery group (38.7 ± 1.9 at 10 minutes of anesthesia, 37.6 ± 1.6 during ischemia and 36.7 ± 2.2 after reperfusion). EtCO2 values did not change during experiment, suggesting that neither surgery nor ischemia induced any acido-basis disorder.

One animal died during ischemia and one died after about 3 hours of reperfusion. Mortaliy in this procedure was therefore 9.1% (2/22). Concerning other animals, measured postoperative pain scores remained very low throughout the postoperative period, and no detectable pain sign was observed beyond 5–6 hours postoperative.

## Efficiency of atipamezole on anesthesia settings

Reversibility of anesthesia allows a quick awakening and a faster and better post-operative recovery of animals. We first tested how effective atipamezole was on our anesthesia parameters (**Fig 3D**). One group of mice anesthetized with alfaxalone/medetomidine/buprenorphine and receiving an atipamezole injection was compared to the group of mice anesthetized with alfaxalone/medetomidine/buprenorphine without anesthesia reversal.

The recovery time drastically reduced after atipamezole administration (20 {18–30} min, compared to 55 {40–70} min in group with no atipamezole, p = 0.000049). Realization of the reflex test did not allow to detect awakening below 10 minutes. In addition, no negative effects suggesting toxicity (diarrhea, muscular tremors, etc.) were observed during post-anesthesia monitoring (24h).

Our data confirmed that addition of atipamezole allowed a quick and safe awakening. We next studied potential confounding effects of the anesthetic protocol on infarction size in our cardiac ischemia-reperfusion model.

### Side effects of the general anesthesia protocol and anesthesia reversal on the infarction size

We investigated if the rapid awakening induced by atipamezole modified the infarction size.

One group of mice was 1) anesthetized with alfaxalone/medetomidine/Buprenorphine, 2) submitted to cardiac ischemia-reperfusion surgery and 3) injected with atipamezole. This group was compared to the group of mice anesthetized with alfaxalone/medetomidine/buprenorphine and submitted to cardiac ischemia-reperfusion surgery.

No significant difference of infarction size normalized by the area-at-risk could be figured out among (Fig 4).

Mean value of Area of Necrosis was 59 ± 6% of Area at Risk in group without atipamezole and 56 ± 6% in group with atipamezole.

There were no significant differences in body weight and proportion of the left ventricle subjected to ischemia (area-at-risk) between the studied groups.

Those data suggested that selected general anesthesia protocol (alfaxalone/medetomidine/ buprenorphine) and reversal by atipamezole did not induce any cardioprotective effect, compared to alfaxalone/medetomidine/buprenorphine group.

## Discussion

The first aim of this study was to test different known anesthetic protocols in order to choose one that could be compatible with the model of murine cardiac ischemia-reperfusion model.

Nowadays, ketamine/xylazine is one of the most common anesthetic protocol (injected intraperitoneal) whereas both the quality of the induced anesthesia and its potential side effects remain controversial [4]. For example, Arras et al. reported that administration of ketamine/ xylazine mixture (100/20 mg/kg) provided only light anesthesia, insufficient for surgery, while higher dosage (150:/0 mg/kg) caused high mortality (40%) [19]. On the other side, Kawai et al. reported that ketamine/xylazine injection (80/8 mg/ml) induced a short anesthesia sufficient for surgery (about 20 minutes) while Siriarchavatana and al. showed that ketamine/xylazine injection (80/10 mg/kg) allowed about 40 minutes of surgical anesthesia [3,17]. In our study, ketamine/xylazine (100/5 mg.kg$^{-1}$) led to about 50 minutes of surgical anesthesia but 2 mice did not reach it. Moreover, it has been showed that ketamine/xylazine anesthesia induces a protective confounding side effect in a rat model of cardiac ischemia-reperfusion by decreasing the infarction size (after 30/120 minutes of ischemia/reperfusion sequence) compared to pentobarbital anesthesia [22]. Altogether, these studies emphasize that anesthetic/analgesic mixture should be studied and selected carefully.

Several anesthetic agents such as alfaxalone and midazolam have been associated with cardiorespiratory depression that could sometimes cause death of the anesthetized animals [23,24]. Nevertheless, we decided to stop alfaxalone/midazolam group for ethical considerations. In this study, such mortality has been drastically improved in alfaxalone/medetomidine protocol, by increasing $O_2$ supply by 30% (0/12 dead mice with 30% $O_2$ supply in this study vs. 2/2 dead mice with 21% O2 supply in previous study). Animals were immediately intubated and ventilated with 30% O2 after the righting was lost in our LAD occlusion model to help avoid those respiratory depression problems.

Finally, among all anesthetic protocols tested in our study, ketamine/medetomidine and alfaxalone/medetomidine protocols were the most noteworthy for the transitory LAD

occlusion model because the surgical anesthesia time is close to 60 minutes (**Fig 2**). Other tested protocols should thus be restricted to short-term surgery (<15-30min) or would need re-injection to allow longer-term surgery. It must be noted that prolonging a liquid anesthesia with re-injection requires appropriate monitoring to correctly target the time for additional injections that can moreover easily lead to fatal overdoses.

In the literature, the efficacy of ketamine/medetomidine protocol on anesthesia is being questioned. For example, it has been reported that with intraperitoneal ketamine/medetomidine administration (100/1 mg.kg$^{-1}$ and 100/5 mg.kg$^{-1}$), none of the animals reached surgical tolerance, despite of a long duration of anesthesia [19] and ketamine/medetomidine anesthesia (75/1 mg.kg$^{-1}$) induces only light anesthesia in rats [20]. Moreover, for ethical considerations, ketamine/medetomidine is usually recommended for animal immobilization in light procedures such as chemical restraint or retroorbital blood collection, with conjunction of atipamezole to reverse medetomidine effects and accelerate recovery [25,26]. The administration of analgesic agents has been reported to modify the quality of anesthesia. Indeed, Bauer et al. showed that pedal withdrawal reaction was lost in all mice injected with butorphanol + ketamine/medetomidine but it was only partially suppressed in the control ketamine/medetomidine group [27].

Alfaxalone/medetomidine protocol (80/0.3 mg.kg$^{-1}$) induced an adequate surgical anesthesia allowing the surgery in our cardiac ischemia-reperfusion model. Our result confirmed other studies reporting long and deep anesthesia using the same or nearby dosage of alfaxalone/medetomidine associated with butorphanol in mice [28,29]. The quality of surgical anesthesia with alfaxalone/medetomidine protocol have also been reported in other animal species [30,31].

The thoracotomy requires a continuous analgesic coverage during and after surgery. Administration of an opioid was thus required to complete the weak analgesic power of ketamine, medetomidine or xylazine. Buprenorphine has been reported to maintain the analgesia for a longer period of time (6-8h) than butorphanol (1-2h [32]). We demonstrated that the substitution of butorphanol by buprenorphine did not interfere with both depth and quality of the anesthesia, whatever alfaxalone/medetomidine and ketamine/medetomidine protocol were chosen (**Fig 3A and 3B**).

All this data prompted us to select alfaxalone/medetomidine, in association with buprenorphine, in our transitory LAD occlusion model. Physiological monitoring (**Table 3**) and time-related parameters of anesthesia (**Fig 3C**) in surgery conditions confirmed that this protocol is suitable for performing long surgical procedures such as cardiac ischemia reperfusion. Although cardiorespiratory depressive effects of alfaxalone, no respiratory failure occured in our hands, surely thanks to intubation and ventilation with 30% O2-enriched air [23,33]. The mean infarction measured after a 60-minutes transitory LAD occlusion in mice anesthetized with alfaxalone/medetomidine/buprenorphine, with or without atipamezole (**Fig 4**), was similar to the one previously obtained in mice anesthetized with fentanyl citrate or pentobarbital (59 ± 6% and 56 ± 6% of AR respectively in this study versus 51 ± 2 and 58 ± 5% in control groups of previous studies) [18,34]. This suggests an absence of interferences between alfaxalone/medetomidine/buprenorphine medication and ischemia-reperfusion mechanisms. Moreover, peri- or post-operative mortality was similar (less than 10%), essentially attributed to either a massive ischemia or a pneumothorax related to the chest muscle suture.

In this study, measured EtCO2, expressed in mmHg, are much lower than expected theoretical EtCO2 (about 17–18 mmHg vs 30–40 mmHg respectively). This difference would be explained by the high respiratory frequency, low tidal volume and the distance between capnograph and mouse, leading together to decrease quantity of expired CO2 detected by capnograph. However, variation of this relative expired CO2 value remains sufficient to highlight a

respiratory problem. No significant variation was measured during the surgery, suggesting that selected anesthesia protocol (alfaxalone/medetomidine/buprenorphine) is suitable for this thoracic surgery in terms of impact on respiratory function.

General and local anesthesia is well known to disrupt thermoregulation mechanisms, causing a mild to severe hypothermia. In this study, we showed that heating pad temperature, which helps to maintain body temperature during anesthesia, decreased significantly among the time. This data seems coherent because concentration as well as effects of injected anesthetic drugs also a fortiori decreases over time.

A study showed that anesthesia reversal by atipamezole modified the animal recovery [35]. In the literature, dose of atipamezole injected to efficiently antagonize medetomidine effects commonly varies from one to five times the initial dose of injected medetomidine [13,35,36]. The data suggest that efficiency of reversal would depend of dose, injection timing, animal species, and anesthetics associated with medetomidine. It has thus reported that antagonist effects of atipamezole (1.5mg/kg) was stronger than atipamezole (0.3mg/kg) when injected 10min after anesthesia induction with medetomidine (0.3mg/kg) whereas both atipamezole doses worked equally when injected 30min after anesthesia induction. We showed here that atipamezole (0.3 mg/kg) administration 70 minutes after anesthesia induction including medetomidine (0.3 mg/kg) allowed a proper recovery (less than 30 minutes after injection. **Fig 3D**) with no adverse reaction (no death, no muscular tremors, and no detectable intestinal disorder was observed during post-operative monitoring). Finally, no significant modification in infarction size was observed when atipamezole was injected few minutes after reperfusion.

In conclusion, we found that in our experimental conditions, the combination of alfaxalone/medetomidine/buprenorphine (80/0.3/0.075 mg.kg$^{-1}$, s.c), associated with reversal by atipamezole (0.3 mg.kg$^{-1}$, s.c., administered 70 minutes after anesthesia induction), was a reliable anesthetic protocol for surgery in male C57/bl6J mice, particularly for the study of ischemia-reperfusion. The main limitation of the study being its restriction to a single sex, it would now be interesting to generalize it with females, especially in the case of preclinical studies. In addition, it has recently been shown that prevention of hypothermia induced by medetomidine/midazolam/butorphanol anesthesia was improved by increasing dose of atipamezole and decreasing dose of medetomidine [37]. These data suggest that optimizing atipamezole and medetomidine doses in the anesthetic protocol we selected could further improve post-operative wakening and animal welfare. To go further it would be relevant to continue the study by also including volatile anesthetics due to their reliability and ability to achieve both rapid induction and recovery [7] and to verify if they are cardioprotective.

## Supporting information

**S1 Fig.**
(PDF)

**S2 Fig.**
(PDF)

**S3 Fig.**
(PDF)

**S4 Fig.**
(PDF)

**S5 Fig.**
(PDF)

**S1 Table.**
(PDF)

## Acknowledgments

We thank Pr. Jean-Claude Desfontis (Ecole Nationale Vétérinaire, Agroalimentaire et de l'Alimentation: Oniris, Nantes, France) and Pr. Karine Portier (Ecole Nationale Vétérinaire de Lyon: VetagroSup, Lyon, France) for accepted to give us an informed opinion on our initial experimental design.

We acknowledge the contribution of the CELPHEDIA Infrastructure (http://www.celphedia.eu/), and especially the technical skills and assistance of the iXplora platform of Lyon-FR.

All authors have read the journal's authorship agreement and policy on disclosure of potential conflicts of interest.

## AI disclosure

The authors did not use generative AI or AI-assisted technologies in the development of this manuscript.

## Author Contributions

**Conceptualization:** Christelle Leon, Lionel Augeul, Catherine Vogt, Bruno Pillot.

**Data curation:** Christelle Leon, Alice Ruelle, Bruno Pillot.

**Formal analysis:** Christelle Leon, Alice Ruelle, Juliette Geoffray, Bruno Pillot.

**Funding acquisition:** Michel Ovize.

**Investigation:** Christelle Leon, Bruno Pillot.

**Methodology:** Christelle Leon, Lionel Augeul, Catherine Vogt, Bruno Pillot.

**Project administration:** Christelle Leon, Bruno Pillot.

**Resources:** Christelle Leon, Michel Ovize, Gabriel Bidaux, Bruno Pillot.

**Software:** Christelle Leon, Bruno Pillot.

**Supervision:** Bruno Pillot.

**Validation:** Christelle Leon, Alice Ruelle, Juliette Geoffray, Lionel Augeul, Catherine Vogt, Pascal Chiari, Michel Ovize, Gabriel Bidaux, Bruno Pillot.

**Visualization:** Christelle Leon, Alice Ruelle, Juliette Geoffray, Lionel Augeul, Pascal Chiari, Ludovic Gomez, Michel Ovize, Gabriel Bidaux, Bruno Pillot.

**Writing – original draft:** Christelle Leon, Bruno Pillot.

**Writing – review & editing:** Christelle Leon, Juliette Geoffray, Lionel Augeul, Catherine Vogt, Pascal Chiari, Gabriel Bidaux, Bruno Pillot.

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
