## [Decision Letter · Decision Letter 0]

7 Jun 2024

PONE-D-24-13742Evaluation of general anesthesia protocols for a highly controlled myocardial infarction model in mice.PLOS ONE

Dear Dr. PILLOT,

Thank you for submitting your manuscript to PLOS ONE. After careful consideration, we feel that it has merit but does not fully meet PLOS ONE’s publication criteria as it currently stands. Therefore, we invite you to submit a revised version of the manuscript that addresses the points raised during the review process.

We look forward to receiving your revised manuscript.

Kind regards,

Carlos Alberto Antunes Viegas, DVM; MSc; PhD

Academic Editor

PLOS ONE

“This work was supported by solely from institutional sources: Université Claude Bernard Lyon 1 and Institut National de la Santé et de la Recherche Médicale (INSERM U1060)”

Reviewers' comments:

Reviewer's Responses to Questions

**Comments to the Author**

1. Is the manuscript technically sound, and do the data support the conclusions?

Reviewer #1: Partly

Reviewer #2: No

2. Has the statistical analysis been performed appropriately and rigorously? 

Reviewer #1: Yes

Reviewer #2: No

3. Have the authors made all data underlying the findings in their manuscript fully available?

Reviewer #1: No

Reviewer #2: No

4. Is the manuscript presented in an intelligible fashion and written in standard English?

Reviewer #1: Yes

Reviewer #2: No

5. Review Comments to the Author

Reviewer #1: I would like to extend my sincere appreciation to the authors for their insightful contribution titled "Evaluation of general anesthesia protocols for a highly controlled myocardial infarction model in mice"

I have some comments and queries regarding specific aspects of the manuscript that I believe would enhance its clarity and impact.

1- Could you explain the reason for selecting buprenorphine and butorphanol as analgesics? Were there any specific considerations regarding their efficacy or potential side effects?

2- Concerning the anesthetic protocol, choosing to antagonize the medetomidine action is relevant and improves the recovery of the animal. However, as analgesia is supported by medetomidine (central analgesia) and butorphanol (slight visceral analgesia), the reversal of the action of medetomidine abolishes its analgesic action and exposes the rat to pain. Why the authors do not use local anesthetics such as lidocaine that would bring a strong peripheral analgesia? Moreover, postoperative analgesia is very light and no data are given concerning the presence and intensity of pain during the postoperative period. The authors have to present data concerning pain assessment in the animals models.

3- Did you observe any differences in post-operative pain scores or recovery profiles among the experimental groups?

4- Are there any insights into the underlying pharmacological mechanisms that might explain why the alfaxalone/medetomidine/buprenorphine protocol was superior for mice cardiac surgery? For example, inflammation that could be relevant to ischemia-reperfusion injury.

5- In the manuscript and title, both 'myocardial infarction' and 'ischemia-reperfusion model' are used to refer to the experimental model. To ensure consistency throughout the text, would it be appropriate to standardize the terminology by selecting one term to describe the model? (ischemia-reperfusion model)

6- Could the authors consider including representative images of triphenyltetrazolium chloride (TTC) staining from each experimental group to visually demonstrate the evaluation of infarct size in the operated mice? The addition of such images would enhance the clarity of the results section.

7- Has the potential impact of the anesthetic protocols on respiratory function been considered, especially given that certain anesthetic agents utilized in the study have been associated with respiratory depression?

8- The general conclusion and evaluation methods used in the present study appear to closely resemble those described in a previously published article by Higuchi et al. ('Evaluation of a combination of alfaxalone with medetomidine and butorphanol for inducing surgical anesthesia in laboratory mice,' Japanese Journal of Veterinary Research, 2016). Could the authors provide clarification and about the novelty of the current research.

Reviewer #2: Dear Authors, the manuscript has the potential to contribute to scientific knowledge on the topic, but requires several modifications to be clear and understandable. There is various information omitted which makes reading rather difficult, not very fluent and at times confusing.

The introduction is not very specific on why there was a real need for this study, and on why it is important to study models of infarction and ischemia. Furthermore, it refers to different species and with different metabolisms.

Materials and methods are unclear. The description is confusing and difficult to reproduce. The division into groups is not clear, the number of animals is not clearly defined and various data regarding monitoring systems are not reported. Furthermore, the data that are declared (ECG, EtCO2, etc.) are then reported only for one group. The ventilation system is not clear (first we talk about intubation, then about mask). It is not clear the criterion with which two groups were subjected to buprenorphine administration, and it is not clear whether butorphanol was also administered in the other groups (this can be seen from a table, but not from the text).

There is no system for evaluating the degree of ischemia.

In the results, there is no clear visualization of the parameters evaluated and the duration of the different phases, as the figures shown have a Y axis with too wide a range.

The discussion part is relatively reduced.

Below are the specific comments.

Major comments :

- Line 72. What is meant by “ immobilization time”?

- Line 97. What is reported is certainly true, but the rabbit is not a rodent and the focus of the study seems to be on rodents, particularly mice. Perhaps it would be more appropriate to speak generically about the cardiovascular effects of drugs without referring to a particular species, or referring specifically to mice, since the response to drugs can vary greatly depending on the species.

- Line 119. Add references for the studies cited on the role of estrogens.

- Lines 132-134. So, how many animals were needed? How many were involved and in which groups? It should already be reported here, even if explained in detail later. Furthermore, it is not further specified how the use of butorphanol and buprenorphine was chosen, because from Figure 1 it seems that all animals received butorphanol . The groups should each be indicated with an acronym for clarity, and with the number of animals for each group.

- Line 136. This entire section is unclear and difficult to repeat. The administration areas, containment, type of gaseous circuit used, etc... are missing.

- Line 137. How were the drugs diluted? It would be useful for the reader who wants to replicate the procedure.

- Table 1. The dose of atipamezole is the same as that of medetomidine. Why this choice? Normally the dose of atipamezole is 2.5-5 times that of medetomidine .

- Table 1. Why were ketamine and xylazine injected intraperitoneally, while all other drugs were injected subcutaneously? Ketamine and xylazine also have good subcutaneous absorption.

- Line 140-141. Here too, it would be useful to add a reference to justify this concept.

- Line 143. How did the physical restraint for the injection occur? Were special tubes used? In which area was the subcutaneous injection performed?

- Line 146. Which gas circuit was used? Was a mask used? Were they intubated? If yes, with what procedure?

- Line 147. So, the non-reversal group underwent the procedures before the others? Also, which group did not receive atipamezole ?

- Line 152 and following. I think the city of the manufacturer is also necessary .

- Line 168. These are not anesthesia parameters but more than anything else they are the evaluation of some reflexes. I would suggest changing the paragraph title.

- Table 2. How was the air blown on the mice eyes to test the reflex?

- Table 2. What is meant by “time” in the table, when evaluating reflexes? Do you intend to talk about the anesthesia plan? Time is a word that may not be clear in this context.

- Line 177. Here is a repetition of what was said before. I suggest reporting the information in just one point, and possibly specifying in which paragraph it will be found.

- Line 177. How were they intubated? With what technique and what tubes? What ventilation settings (pressure, acts, etc.)?

- Line 179. How were these parameters monitored? With what tools?

- Line 179. It is not the system that is placed on the tail, but the sleeve. The sentence should be written better.

- Line 180. What is meant by “ immobilization time”? when using these terms, specific definitions should be given to rely on.

- Line 181. Better indicate the area where the needles were inserted and how they were connected to the ECG system. Indicate the name of the monitoring system. Also, “on” is not appropriate if the insertion of an instrument into the subcutaneous tissue is involved.

- Line 185. What suture pattern?

- Line 188. Indicate the name and details of the manufacturer for the incubator.

- Line 188. What is meant by “ total recovery”? give a definition. Furthermore, following the meaning of the sentence, we understand that the food and water were left only until recovery, and then removed. Is that so?

- Lines 189. Although the focus is on anesthesia, the methods should also be briefly described here.

- Line 193. Since nonparametric tests were performed, it is assumed that the distribution was not considered normal based on the Shapiro-Wilk test. Therefore, the use of median and interquartile ranges for describing data would be more appropriate than the use of mean and SD.

- Line 198. What “times” are you referring to? It has not been specified anywhere previously in the text, except perhaps in Table 2. However, Table 2 is not at all clear and there is no adequate definition of what is meant. Furthermore, the evaluation of reflexes can allow us to hypothesize a specific condition, rather than a "time". I believe that the reasoning behind this analysis should be explained better because it is not clear at the moment. Furthermore, in the "Statistical analysis " section it would be more appropriate to clearly talk about which tests were done on which parameters specifically.

- Line 210. From Figure 1 it is not at all clear that LRR was lost in less than 3 minutes, being a figure with a Y axis ranging from 0 to 200. The 3 minute point is imperceptible. I suggest modifying the figure, and possibly separating it.

- Line 215. So, in these 2 animals, the procedure was not performed?

- Line 220-221. So, was the duration of surgical anesthesia compared with statistical tests? This must be specified in the statistical analysis part which, as mentioned, is not clear.

- Line 223. According to what was written before, the animals had been intubated. why are we talking about face masks here? Also, which mask? What type of ventilation?

- Line 224. On what basis is it presumed to be “ potentially due to cardiorespiratory failures ”? Was there apnea? How did the heart rate change? What was done with the 2 untested mice, given that we stopped at 4? The total number declared in the abstract is 72. The number is not included in the text, and this is lacking information.

- Line 228. According to the reported route of administration, ketamine is administered intraperitoneally while medetomidine is administered subcutaneously. Therefore, it is not a “single injection”.

- Line 231. Since ketamine , medetomidine and xylazine also have analgesic power, unlike alfaxalone and midazolam , the role of analgesia is already partly important in the evaluation of protocols without the use of opioids. Thus, the role of analgesia was not evaluated, but the role of adding an opioid to the protocol. I suggest changing the title of the paragraph and considering this aspect in the discussion.

- Line 233. From Figure 1, it appears that all animals in the previous evaluation received butorphanol , but this was never specified before. Is this an error in the table? However, if it has been administered to everyone, the combination of drugs itself must be analyzed considering the effects of the opioid.

- Line 234 and following. Was this assessment done in new groups? Or are they the same groups we were talking about before? If they are new groups, specify. On how many animals was the procedure carried out? However, if we are talking about animals in the groups previously described, the use of opioids certainly changes the assessment made. It is not clear which animals and which groups we are talking about, and whether these evaluations were carried out only after the first analyzes had already been carried out, on the protocols considered to be the best.

- Line 236. How were the comparisons made? With what test? Are there any significant results or not? If a statistical analysis has not been performed, I believe it should be implemented instead.

- Line 243. How time is determined is “ significantly higher ”? indicate a p value .

- Line 250. So, the procedure was performed in this group only? If so, the paragraph at line 174 should be partially moved here. Why was it placed before?

- Line 259. Is the EtCO2 value reported in the other groups? In what table or figure? What does figure 2C have to do with it?

- Line 261. Normally, " immobilization " means a phase in which the animal is open to manipulation, immediately after the administration of anesthetics. What is meant by “ immobilization ” in the context of the recovery phase? It seems counterintuitive.

- Line 263-264. We talk about animals that have undergone surgery and animals that have not undergone it, and then 3 differences in times are reported. It is not clear which group is being referred to. Indicate.

- Line 266. Systolic blood pressure.

- Table 3. The table description is confusing. The table should be set up differently, in order to insert the IP value of the comparisons made, so that they are clear.

- Table 3. Why EtCO2 is expressed as “ arbitrary unit ”? normally we talk about mmHg.

- Line 289. Body temperature was maintained, but there was a difference in pad temperature . Why? Since the parameter has been recorded, it would be appropriate to discuss it briefly.

- Lines 289-290. EtCO2 certainly reflects blood pCO2, but its variation can depend on various factors. Furthermore, the reported values are rather low to be considered normal: again, the monitoring system can influence this analysis, as the tidal volume is very low in this species. However, these aspects require a brief discussion. Furthermore, during the ischemia phase there was an increase, as expected.

- Line 295. Here too, it is not well defined how many subjects were tested, on which groups, etc.

- Line 297. Is the recovery time assessment based on what is reported in Table 2? So, was it enough to obtain a score between 4 and 1 to consider the animal awake? It is a very wide range, which can be due to several factors. Not very indicative of an awakening. Does it mean the recovery of the righting reflex instead?

- atipamezole was not administered , from what moment was the evaluation carried out? In these cases, time should be considered from a specific point valid for both groups.

- Lines 300-301. Were there cases in which monitoring was not possible because they were already awake? How many?

- Line 308. This procedure and following evaluation and analysis need to be better described and supported by statistical analysis, and p values should be reported.

- Line 333. It was never reported, in the Materials and methods section, the use of 21% oxygen, but only the use of 30% oxygen.

- Line 333-334. Intubation alone does not solve respiratory problems if controlled ventilation with a set percentage of oxygen is not carried out. The phrase is a bit misleading.

- Line 337. The duration of the procedure should be reported in the results.

- Line 351. If the doses are the same, why was this investigation necessary?

- Line 354. The surgical plane of anesthesia can differ among species, so the evaluation of reflexes is not necessarily an indicator of surgical plane.

- Line 365. It is stated the infarction size was similar to other studies, but no data is reported, so it is not useful for the reader.

- Line 369. Did you have 10% mortality? Where is this declared?

- Line 370. Was a negative pressure instituted to resolve pneumothorax after the procedures? This must be discussed in the Materials and methods section.

- Line 372. Why should 70 minutes be considered as a late administration? I would remove this adjective.

- Line 373. “ less than 10 minutes”. In the results, you state that the recovery was 22 ± 9 min.

- Line 378 “ alfaxalone+medetomidine+buprenorphine (80/0,3/1 mg.kg -1 , sc )”. In Table 1, the dose of buprenorphine is 0.075 mg/kg. Also, this error is found in the abstract.

- Line 379. It was never stated that a “variable dosage” was used. What is meant by this phrase? Why is it not reported in the materials and methods and in the results?

- Lines 377-380. In the materials and methods, it is stated that the evaluation of the surgical procedure was made on the protocol based on alfaxalone and medetomidine . Buprenorphine is not mentioned. While reading, there is confusion.

Minor comments :

- Throughout the text. “infarct” -> “infaction”

- Line 34. I would change the running Head to: “Mice anesthesia for myocardial infaction model”, so that it is more specific.

- Line 38. “alfaxalone” instead of “ alfaxan ”

- Line 62. Better to replace “mouse” with “mice”.

- Line 71. “ did not induce major change in” -> “did not alter”

- Line 82. “provide” -> “ provides ”

- Line 82. Remove “a”.

- Line 108. Reversal could be a better term.

- Line 110. Commas are not needed.

- Line 124. “ was ” instead of “ were ”, in both cases.

- Line 124. Remove the bracket.

- Line 144. “placed in a dorsal recumbency”: remove “a”.

- Line 146. Same as line 144.

- Line 146. “was” instead of “were”.

- Line 152. “ alcyon ” -> “ Alcyon ”.

- Table 2. “ refex ” -> “reflex”

- Line 178. Remove the comma after “body”.

- Line 179. Non-invasive.

- Line 182. “ achieved ” -> “ performed ”

- Line 183. A dot is missing.

- Line 184. “loosened” -> “loosening”.

- Line 186. “was” instead of “were”.

- Line 188. Add an article before “ heating incubator”.

- Line 193. Results are expressed .

- Line 218. Surgical instead of surgery.

- Results section . Sometimes it is read “ mn ”, sometimes “min”. follow the journal’s style of abbreviations.

- Line 229. “ adding ” -> “ subsequently administering ”

- Line 230. “ awakening ” -> “recovery”

- Line 274. Remove “ were ”.

- Line 286. “did” instead of “is”.

- Line 299. I suggest “reduced” instead of “fell”.

- Line 301. Effects.

- Line 302. Were observed.

- Line 323. Led.

- Line 330. “That” instead of “what”.

- Line 333. A dot is missing.

- Line 341. “ we can read” -> “it has been reported”.

- Line 343. “ sleeping time” -> “duration of anesthesia”.

- Line 344. Induces.

- Line 351. I suggest “The quality of surgical anesthesia…”

- Line 354. Surgical.

- Line 357. Remove “;”.

- Line 365. Was.

- Line 368. Peri-.

Best regards

The Reviewer

6. PLOS authors have the option to publish the peer review history of their article (what does this mean?). If published, this will include your full peer review and any attached files.

Reviewer #1: **Yes: **Ahmed Farag

Reviewer #2: No

---

## [Author Response · Author response to Decision Letter 0]

22 Jul 2024

First, we would like to thank the editor and the reviewers for considering our paper. Thanks for the reviewing : questions, comments, tips and corrections. 

We have modified the manuscript and figures/tables as required (corrections, clarifications, informations adding...) and hope to have properly answered to all your questions. 

Answers to every points raised have been compiled in separated file

We thank you in advance for the consideration of this revised version of our manuscript

---

## [Decision Letter · Decision Letter 1]

20 Aug 2024

Evaluation of general anesthesia protocols for a highly controlled cardiac ischemia-reperfusion model in mice.

PONE-D-24-13742R1

Dear Dr. Bruno Pillot,

We’re pleased to inform you that your manuscript has been judged scientifically suitable for publication and will be formally accepted for publication once it meets all outstanding technical requirements.

Kind regards,

Carlos Alberto Antunes Viegas, DVM; MSc; PhD

Academic Editor

PLOS ONE

Additional Editor Comments (optional):

Reviewers' comments:

Reviewer's Responses to Questions

**Comments to the Author**

1. If the authors have adequately addressed your comments raised in a previous round of review and you feel that this manuscript is now acceptable for publication, you may indicate that here to bypass the “Comments to the Author” section, enter your conflict of interest statement in the “Confidential to Editor” section, and submit your "Accept" recommendation.

Reviewer #1: (No Response)

Reviewer #3: All comments have been addressed

2. Is the manuscript technically sound, and do the data support the conclusions?

Reviewer #1: (No Response)

Reviewer #3: Yes

3. Has the statistical analysis been performed appropriately and rigorously? 

Reviewer #1: (No Response)

Reviewer #3: Yes

4. Have the authors made all data underlying the findings in their manuscript fully available?

Reviewer #1: (No Response)

Reviewer #3: Yes

5. Is the manuscript presented in an intelligible fashion and written in standard English?

Reviewer #1: (No Response)

Reviewer #3: Yes

6. Review Comments to the Author

Reviewer #1: (No Response)

Reviewer #3: Accept submission. Accept submission. Accept submission. Accept submission. Accept submission. Accept submission.

7. PLOS authors have the option to publish the peer review history of their article (what does this mean?). If published, this will include your full peer review and any attached files.

Reviewer #1: No

Reviewer #3: No

---

## [Editor Report · Acceptance letter]

27 Aug 2024

PONE-D-24-13742R1 

PLOS ONE

Dear Dr. Pillot, 

I'm pleased to inform you that your manuscript has been deemed suitable for publication in PLOS ONE. Congratulations! Your manuscript is now being handed over to our production team.

Kind regards, 

on behalf of

Dr. Carlos Alberto Antunes Viegas 

Academic Editor

PLOS ONE